# Noticeable Quantities of Functional Compounds and Antioxidant Activities Remain after Cooking of Colored Fleshed Potatoes Native from Southern Chile

**DOI:** 10.3390/molecules26020314

**Published:** 2021-01-09

**Authors:** Stefano Ercoli, José Parada, Luis Bustamante, Isidro Hermosín-Gutiérrez, Boris Contreras, Pablo Cornejo, Antonieta Ruiz

**Affiliations:** 1Departamento de Ciencias Químicas y Recursos Naturales, Scientific and Technological Bioresource Nucleus BIOREN-UFRO, Universidad de La Frontera, Avda. Francisco Salazar 01145, 4811230 Temuco, Chile; ercoli.95@gmail.com (S.E.); j.parada03@ufromail.cl (J.P.); pablo.cornejo@ufrontera.cl (P.C.); 2Centro de Investigación en Micorrizas y Sustentabilidad Agroambiental CIMYSA, Universidad de La Frontera, 4811230 Temuco, Chile; 3Departamento de Análisis Instrumental, Facultad de Farmacia, Universidad de Concepción, P.O. Box 160-C, 4030000 Concepción, Chile; lbustamante@udec.cl; 4Instituto Regional de Investigación Científica Aplicada, Universidad de Castilla–La Mancha, Av. Camilo José Cela s/n, 13071 Ciudad Real, Spain; isidro.hermosin@uclm.es; 5Novaseed Ltda. and Papas Arcoiris Ltda., Loteo Pozo de Ripio s/n, Parque Ivian II, 5550000 Puerto Varas, Chile; boriscontreras@novaseed.cl

**Keywords:** colored-fleshed potato, anthocyanins, cooking, antioxidant activity, HPLC-DAD-ESI-MS/MS, antioxidants

## Abstract

The effect of cooking on the concentrations of phenolic compounds and antioxidant activities in 33 colored-fleshed potatoes genotypes was evaluated. The phenolic profiles, concentrations, and antioxidant activity were evaluated with a liquid chromatography diode array detector coupled to a mass spectrometer with an electrospray ionization interface (HPLC-DAD-ESI-MS/MS). Eleven anthocyanins were detected; in the case of red-fleshed genotypes, these were mainly acyl-glycosides derivatives of pelargonidin, whereas, in purple-fleshed genotypes, acyl-glycosides derivatives of petunidin were the most important. In the case of the purple-fleshed genotypes, the most important compound was petunidin-3-coumaroylrutinoside-5-glucoside. Concentrations of total anthocyanins varied between 1.21 g kg^−1^ in fresh and 1.05 g kg^−1^ in cooked potato and the decreases due to cooking ranged between 3% and 59%. The genotypes that showed the highest levels of total phenols also presented the highest levels of antioxidant activity. These results are of relevance because they suggest anthocyanins are important contributors to the antioxidant activity of these potato genotypes, which is significant even after the drastic process of cooking.

## 1. Introduction

Potato (*Solanum tuberosum* L.) is one of the most important food crops worldwide due to its higher consumption and gastronomic value, containing important concentrations of polyphenols in the same magnitude order as coffee, berries like grapes or blueberries, and seeds, such as almonds [1]. Southern Chile, specifically Chiloé Island, has been described as one of the main genetic reservoirs of a great diversity of local ecotypes of white- and colored-fleshed potatoes that currently have been described mainly from an agronomic perspective. Additionally, the consumption of colored-fleshed potatoes is lower because consumers are less aware of the existence of these colored genotypes and the potential beneficial health effect from their compounds. Therefore, these potatoes are mainly consumed in gourmet restaurants. There are few studies on the profiles of antioxidants, including phenolic compounds or other bioactive compounds in Chilean potatoes [2,3].

Anthocyanins, phenolic compounds reported mainly in berries, have potential benefits for human health [4,5,6]. For instance, in the case of cranberries, anti-adhesion activity against uropathogenic bacteria has been reported [7]. These compounds have also been reported in some genotypes of vegetables such as black tomato, maize, onions, carrots, red cabbage, and potatoes, being associated with the prevention of diseases and pathologies [7,8,9,10,11]. The antioxidant compositions of colored-fleshed potatoes from other latitudes have been only scarcely reported. In red-fleshed potatoes, acetylated derivatives of pelargonidin were reported as the most abundant compounds [12], while in purple-fleshed potatoes, the acetylated derivatives of malvidin, petunidin, peonidin, and delphinidin were reported [9]. For example, anthocyanins decrease from 7 µg g^−1^ DW to 0.6 µg g^−1^ after cooking and 0.1 µg g^−1^ after frying processes in Korean potatoes [13], while in potato genotypes from the Czech Republic, total concentrations between 0.2 to 2.4 µg g^−1^ were detected, and these values were influenced by variety and geographical origin [14]. In Latin America, the polyphenolic composition of Andean potatoes has been studied, with levels of total anthocyanins ranging from 0.25 to 2.4 µg g^−1^ DW in raw potatoes and 0.3–1.8 µg g^−1^ DW in cooked potatoes [15]. It is noticeable that pelargonidin, the major anthocyanin reported in colored-fleshed potatoes, and malvidin derivatives, are most stable than derivatives of other anthocyanins, such as petunidin, during the frying process [16]. There is little information about the phenolic composition of Chilean colored-fleshed potatoes, mainly related to the profiles of anthocyanins and hydroxycinnamic acids (HCAD) after frying [2].

Some biological effects have been described for the polyphenolic extracts of colored potatoes and give them beneficial effects towards some cancers [17], as anticarcinogenic effects of anthocyanin extracts, which serve as suppressors of early and advanced cell proliferation and inductors of apoptosis in colon cancer under in vitro and in vivo conditions [18], and also in human leukemia cells [19]. The anthocyanins from purple-fleshed potatoes have also been reported that decreases postprandial glycemic response and affect inflammation markers [20]. Additionally, the anthocyanin fractions of purple-fleshed potatoes have the potential to attenuate the UVB-irradiated skin oxidative stress and inflammatory response [21]. Therefore, it is of high relevance to understand the effects of the cooking process on the antioxidant profiles and the associated activities in colored potatoes in relation to their role in human health. The main objective of the present work is to describe the composition and activities of antioxidant compounds in a wide range of germplasms of colored potatoes developed in Chile and contribute to a more comprehensive description of the potential use of this underexploited food resource.

## 2. Results

### 2.1. Identification of Anthocyanins and Hydroxycinnamic Acids

The anthocyanins in the 33 genotypes, including both fresh and cooked samples were determined using a liquid chromatography diode array detector coupled to a mass spectrometer with an electrospray ionization interface (HPLC-DAD-ESI-MS/MS). These compounds were identified based on the spectroscopic characteristics of each compound (UV-Vis), and the assignments were confirmed using low-resolution MS/MS spectra (Appendix A) and published data complying with a two-level annotation [22]. The results showed a large number of anthocyanins in both the fresh and cooked potatoes (Table 1; Figure 1, Appendix A). Although anthocyanins were not detected in white-fleshed potatoes, in fresh red- and purple-fleshed potatoes, a total of eleven anthocyanins were detected. The compounds were mainly acyl glucoside derivatives of pelargonidin. In purple-fleshed samples, the most important compound was petunidin-3-*p*-coumaroylrutinoside-5-glucoside (petunin) (peak five), characterized by a (de)protonated molecular ion at *m*/*z* 933.2 and fragment ions at *m*/*z* 771.2, 479.0, and 317.0; however, in red-fleshed potatoes, the most important anthocyanin was pelargonidin-3-*p*-coumaroylrutinoside-5-glucoside (peak six). The other detected derivatives were pelargonidin-3-caffeoylrutinoside (peak one), pelargonidin-3-rutinoside (peak two), pelargonidin-3-caffeoylrutinoside-5-glucoside (peak three) and pelargonidin-3-feruloylrutinoside-5-glucoside (peak nine). For all the later compounds, a common fragment ion of *m*/*z* 271.2 was detected, which corresponded to pelargonidin aglycone, and neutral losses of 162 amu, corresponding to the loss of a 5-glucose was observed in the case of the 3,5-diglucoside derivatives. In addition, a minor pelargonidin derivative (peak 11) was also found, and that derivative was characterized by the neutral loss of 190 amu, together with the neutral losses corresponding to the 3-*p*-coumaroylrutinoside moiety. This compound likely corresponds to an artifact produced during the extraction procedure after the formylation of the main anthocyanin in red-fleshed potatoes, and it was tentatively assigned as pelargonidin-3-*p*-coumaroylglucose-5-formylglucose, which was also detected in red-fleshed genotypes.

Noticeably, in purple-fleshed potatoes, derivatives of petunidin were the most important; however, in red-fleshed potatoes, pelargonidin derivatives were mainly detected. Four anthocyanins, 3-coumaroylrutinoside-5-glucoside derivatives were detected mainly in purple varieties, showing neutral losses of 454 amu for the *p*-coumaroylrutinoside group and 162 amu for the glucoside group. A 3-feruloylrutinoside-5-glucoside derivative of petunidin, characterized by a molecular ion of *m*/*z* 963.3 and product ions of *m*/*z* 479.0 and 317.0, was also detected in purple potatoes. The anthocyanins profiles of cooked samples were similar to those in fresh samples of the same varieties, and no newly formed compounds were detected. On the other hand, hydroxycinnamic acids as 3-, 4-, and 5-caffeoylquinic acids were detected in all genotypes.

### 2.2. Quantification of Anthocyanins and Hydroxycinnamic Acids

Anthocyanin quantification was carried out by external calibration using petunidin-3-glucoside as the standard, with the equation y = 92,803x − 13,997, standard error (s_XY_) 4832.37, R^2^ 0.9998, detection limit (DL) 0.16 mg L^−1^, quantification limit (QL) 0.52 mg L^−1^, and a linear range between 0.52 and 40.00 mg L^−1^, where DL was calculated as 3 × s_xy_ slope^−1^ and QL as 10 × s_xy_ slope^−1^. The efficiency of the anthocyanin extraction procedure was evaluated using a potato sample (CB2011-104 genotype), where seven successive extraction steps were carried out, and quantitative results were obtained for each one of them. As a result, after seven steps, no further increase in anthocyanin concentration was detected, obtaining after four steps, 91% and 80% recovery for fresh samples and cooked samples, respectively. The extraction precision expressed as a CV was 10.51% in fresh samples and 6.6% in cooked samples. The concentrations of total anthocyanins were determined as the sum of the concentrations of the individual compounds quantified by high-performance liquid chromatography diode array detection (HPLC-DAD). The levels of total anthocyanins in fresh potatoes varied between no detected to 1.21 g kg^−1^ fresh weight (FW), and the most abundant compound was petunidin-3-coumaroylrutinoside-5-glucoside, which reached levels between 0.01 to 0.9 g kg^−1^ FW in fresh potatoes and 0.001 to 0.85 g kg^−1^ FW in cooked potatoes. In cooked potatoes, the total anthocyanin contents reached up to 1.05 g kg^−1^ FW (Figure 2). In CR2012.363, CB2012.350, CB2012.128, and CB2012.347 genotypes, lower concentrations of anthocyanins were detected in both the fresh and cooked samples (Appendix A).

For HCADs, quantification was carried out using 5-caffeoylquinic acid as the standard, with the equation y = 48,711x − 15,778, R^2^ 0.9997, s_xy_ 12,988.4, DL 0.8 mg L^−1^, QL 2.7 mg L^−1,^ and a linear range between 2.7 and 140.00 mg L^−1^. HCADs were detected at concentrations between 0.06 and 2.0 g kg^−1^ FW in fresh potatoes and between 0.17 and 4.23 g kg^−1^ FW in cooked potatoes, presenting the CB2011-104 genotype the highest total concentration in fresh and cooked potatoes (Appendix A). On the other hand, the most abundant compound was 5-caffeoylquinic acid with 1.96 g kg^−1^ FW in fresh and 3.51 g kg^−1^ FW in cooked potatoes from CB2011-104 genotype. These results are of importance since the concentrations of HCADs are well-related to the antioxidant activity and also with the concentrations of total anthocyanins.

### 2.3. Total Phenols and Antioxidant Activity, before and after Processing

The Folin-Ciocalteau method provides an estimate of the total phenols contained in a sample in terms of the gallic acid equivalents. In fresh potatoes, the total phenols concentrations ranged between 0.2 and 2.4 g kg^−1^ FW, with a mean value of 1.03 g kg^−1^ FW, while after cooking, the concentrations were between 0.2 and 2.3 g kg^−1^ FW with a mean value of 0.7 g kg^−1^ FW (Table 2). Despite the diminution of total phenols after cooking, the concentrations in CB2011.104 (which presented the highest levels of total phenols) decreased by only 3.6% relative to the fresh sample. In contrast, the trolox equivalent antioxidant capacity (TEAC) method is an important methodology for estimating the antioxidant activity in samples with higher concentrations of anthocyanins, for example, berries or other colored foods such as maize. In fresh samples, the antioxidant activities ranged between 0.5 and 8.4 µmol g^−1^ FW with a mean of 2.9 µmol g^−1^ FW. After cooking, this range of concentrations varied between 0 and 8.1 µmol g^−1^ with a mean of 3 µmol g^−1^ FW (Table 2).

## 3. Discussion

Although a petunidin derivative was the most abundant compound detected in the studied purple-fleshed potatoes, in other potatoes from different latitudes, pelargonidin derivatives [23,24], malvidin derivatives [25,26], petunidin and peonidin derivatives [24] or petunidin derivatives [23] were the most important, depending on the variety. Pelargonidin was the anthocyanidin that had the highest number of derivatives present, and 3-rutinoside, 3-caffeoylrutinoside, 3-caffeoylrutinoside-5-glucoside, 3-feruloylrutinoside-5-glucoside, and 3-coumaroylrutinoside-5-glucoside derivatives were the next most abundant of this series with concentrations up to 0.18 g kg^−1^ FW in fresh potatoes and 0.14 g kg^−1^ FW in cooked potatoes in red-fleshed genotypes. On the other hand, in sweet potato (*Ipomoea batata L*.), 3,5-O-dicaffeoylquinic acid, astragalin, and cyanidin have been reported as the most abundant phenolic compounds in leaves and flesh, respectively [27], whereas the concentrations of total anthocyanins are comparable to the reported for colored genotypes from Ecuadorian Andean region were concentrations between 2.74 and 172.53 ug g^−1^ FW had been reported [28]. Although that higher concentrations of the phenolic compound have been detected in potato flesh, it has been reported that approximately 50% of these molecules are located in the peel and adjacent sections, including hydroxycinnamic and hydroxybenzoic acids and also non-anthocyanin flavonoids as flavonols and flavan-3-ols, among others [29,30]

High temperatures can induce thermal degradation of anthocyanins following hydrolytic cleavage of glycosidic bonds and/or cleavage of the flavonoid core [31]. This degradation decreases the anthocyanin concentrations in foods during cooking or storage, and the degradation is accelerated by the presence of oxygen and light [32,33]. This effect has been observed when the samples were subjected to different treatments involving the use of high temperatures in the cooking process (up to 100 °C). Interestingly, our results showed that total anthocyanin concentrations in the potatoes decrease by between 2.8% and 59%, being in almost all cases the reduction significant (Figure 3), but much lower than expected based on previous reports.

On the other hand, the anthocyanins concentrations in some genotypes of colored-fleshed potatoes are higher than the concentrations detected in other foods as berries like murtilla (*Ugni molinae*) and comparable to the levels in blueberries (*Vaccinium myrtillus*) (0.09 g kg^−1^ and 1.13 g kg^−1^, respectively), according to previously described by Ruiz et al. [5]. This aspect is crucial since anthocyanin concentrations are relevant from a nutritional perspective because potato consumption is much higher than berry consumption; thus, the consumption of colored-fleshed potatoes can represent an interesting source of antioxidants with potential beneficial effects for human health. It is noticeable the higher concentrations of HCADs here were detected in colored genotypes, considering that, as previously reported, chlorogenic acids represent over 90% of total phenolics in white potatoes [30] and were even higher in the colored-fleshed potatoes assayed here.

Notably, the samples with the highest levels of total phenols also presented the highest levels of antioxidant activity and anthocyanins contents [34], and high r Pearson coefficients were observed (anthocyanins vs. Folin r = 0.89; anthocyanins vs. TEAC r = 0.87; Folin vs. TEAC r = 0.87). Moreover, the multivariate principal component analysis corroborated this close relationship among variables, and the most colored genotypes (CB2011.104) can be grouped in a specific cluster with high antioxidant activity (TEAC), total phenols (PHENtot), total anthocyanins (ANTtot) and also total hydroxycinnamic acid derivatives (HCADtot), including the most abundant anthocyanin in purple genotypes (ant5) (Figure 3). On the other hand, TR2012.078, a red-colored genotype, can be grouped with anthocyanin 6 (pelargonidin-3-*p*-coumaroylrutinoside-5-glucoside). These results are relevant because an important percentage of the antioxidant activity of potato would be provided by the secondary metabolites as phenolic compounds [2]. Total phenols from colored potatoes analyzed here had a result much higher than the phenolic content in other vegetables such as, as broccoli, spinach, brussel sprouts, among others [30]. On the other hand, potato is generally consumed after cooking and not fresh, and contrary to popular belief, processed foods still can retain important levels of antioxidants that would be provided to the final consumer in important amounts depending on the genotypes selected for consumption. 

Considering potato is a widely consumed vegetable throughout the world, the consumption of colored-fleshed potatoes after cooking must be considered as an important source of antioxidants. It has been previously reported that phenolic compounds like anthocyanins or phenolic acids are unstable under light or high temperatures [31]. Therefore, since cooking uses temperatures of approximately 100 °C, the structure and contents of these secondary metabolites will likely be affected by cooking, probably associated with degradation reactions but also the extractability of antioxidant compounds from the cellular matrix changes during the cooking processes [35] or the liberation of phenolics by hydrolysis of the glycoside bonds during the treatments is possible. However, our results strongly suggest that cooked potatoes still provided valuable levels of phenolic compounds and antioxidant activity to the final consumer independent of the temperature used in their processing.

## 4. Materials and Methods

### 4.1. Samples

Potato samples were obtained from Novaseed Ltda. and Papas Arcoiris Ltda. (Puerto Varas, southern Chile), which have developed lines of uniform native colored potatoes with contrasting characteristics and uniformity. A total of 33 genotypes of potatoes developed by crosses were used, including two white-fleshed genotypes (BWF and VR808), six red-fleshed genotypes (CR2002.8; CR2012.363, TR2012.078; TY2012.365; CB2011.568 and CB2012.361), and 25 purple-fleshed genotypes (Figure 4). All potato genotypes were analyzed in fresh and cooked. Cooked samples were processed in boiling water until the condition of potato was adequate for consumption after punching with a fork.

### 4.2. Identification and Quantification of Anthocyanins and Hydroxycinnamic Acids

Extraction of anthocyanins and hydroxycinnamic acids was done according to Ruiz et al. [2]. Peeled fresh or cooked potato (10 g) was washed with distilled water, sliced in small pieces, homogenized in solvent mixture water/acetonitrile/formic acid (87/3/10; *v/v/v*), sonicated by 10 min, and shaken for 16 h (200 rpm). The supernatant was separated by centrifugation (3000 rpm; 10 min). The above extraction procedure was repeated up to four times, and in the last cycle, was only shaken for 10 min. Finally, the supernatants were mixed, and the extract was stored at −20 °C until analyzed. HPLC-DAD-ESI-MS/MS analyses of the phenolic compounds were carried out using a Shimadzu HPLC system (Tokyo, Japan) equipped with a quaternary LC-20AT pump with a DGU-20A5R degassed unit, a CTO-20A oven, and an UV-vis diode array spectrophotometer (SPD-M20A). The compounds were identified by coupling that HPLC-DAD system to a mass spectrometer (QTrap LC/MS/MS 3200 Applied Biosystem MDS Sciex system, Foster City, CA, USA), using Analyst software (v. 1.5.2) for MS/MS analysis and Lab Solutions for DAD analysis. Phenolic analysis by HPLC was carried out according to Ruiz et al. [4] using a C_18_ column (Kromasil 250 × 4.6 mm, 5 μm) with a C_18_ precolumn (NovaPak, Waters, 22 × 3.9 mm, 4 μm) with a mobile phase gradient of A) water/acetonitrile/formic acid, 87/3/10; *v*/*v*/*v* and B) water/acetonitrile/formic acid, 40/50/10; *v*/*v*/*v*. The mobile phase B gradient was from 6% to 30% in 15 min, from 30% to 50% in 15 min, from 50% to 60% in 5 min, and from 60% to 6% in 6 min, followed by stabilization for 10 min, at a flow rate of 0.8 mL min^−1^ at 40 °C. The anthocyanins identities were identified based on a comparison of their MS/MS spectra with those previously reported in identification studies [25,36], whereas for HCADs, the identifications were also confirmed by comparison with retention times of commercial standards complying with level one, proposed by Sumner et al. [22]. Quantification was carried out at 520 nm for anthocyanins using petunidin-3-glucoside (Phytolab, Vestenbergsgreuth, Germany) as standard and at 320 for hydroxycinnamic acid with 5-caffeoylquinic acid (Sigma-Aldrich, Steimheim, Germany) as standard. Total phenols were determined by the Folin–Ciocalteau method described by Parada et al. [37] with modifications using a Thermo Scientific Genesys 105 UV-vis spectrometer and gallic acid (Sigma-Aldrich, Steimheim, Germany) a as standard, where the reagents were added in the following order: 250 μL of gallic acid or sample, 12.5 mL of water, 1250 μL of Folin-Ciocalteau reagent, 5 mL of sodium carbonate and was completed to a volume of 25 mL with water. Determinations were made at 750 nm.

### 4.3. Determination of Antioxidant Activity

Antioxidant activity was determined using trolox equivalent antioxidant capacity (TEAC). The assay was developed using the extracts obtained from fresh and boiled potatoes in a Thermo Scientific Genesys 105 UV-vis spectrometer, using trolox (a water soluble analog of vitamin E) as a standard [38]. In the plastic cuvette, 1 mL of ABTS radical was added, then the first absorbance reading (734 nm) was done, then 20 μL of trolox curve (0.0–2.0 mmol L^−1^) or sample was added. It was left incubating for 6 min at 30 °C in the dark to finally do a second reading at 734 nm. Antioxidant activity was expressed as Trolox equivalents.

### 4.4. Statistical Analysis

For all the variables, ANOVA was performed to test the effect of the different potato genotypes. Data not meeting with normality and/or homoscedasticity were transformed by Ln, but the results are shown in their original scale. Multiple range Tukey’s test was used to post hoc compare the means of treatments (*p* < 0.05). Factorial analysis using principal component (PC) extraction and also correlation by means of Pearson coefficient among scalar variables were determined. For the comparison of means for total anthocyanins, TEAC and Folin under fresh and cooked conditions, a Student’s *t*-test was used (*p* < 0.05). All the statistics were performed using the IBM SPSS Statistics software v. 23 (IBM Corp.)

## 5. Conclusions

According to our results, colored-fleshed potatoes are a noticeable source of natural antioxidants with interesting characteristics in amount and composition. It is also relevant that the high levels of both antioxidant compounds and activities after cooking ensure high availability to the final colored-potato consumer. The previous is an important advantage that justifies advanced studies of the genotypes used here as functional foods and also adds support for the breeding programs that are currently in progress. This aspect is important for increasing the consumption of these kinds of colored-potatoes, establishing the related genetic improvement programs for selecting those varieties with the best agronomical and functional characteristics, and also creating an interesting alternative for a global increase in the consumption of colored-fleshed potatoes, which is currently scarce.

## Figures and Tables

**Figure 1 molecules-26-00314-f001:**
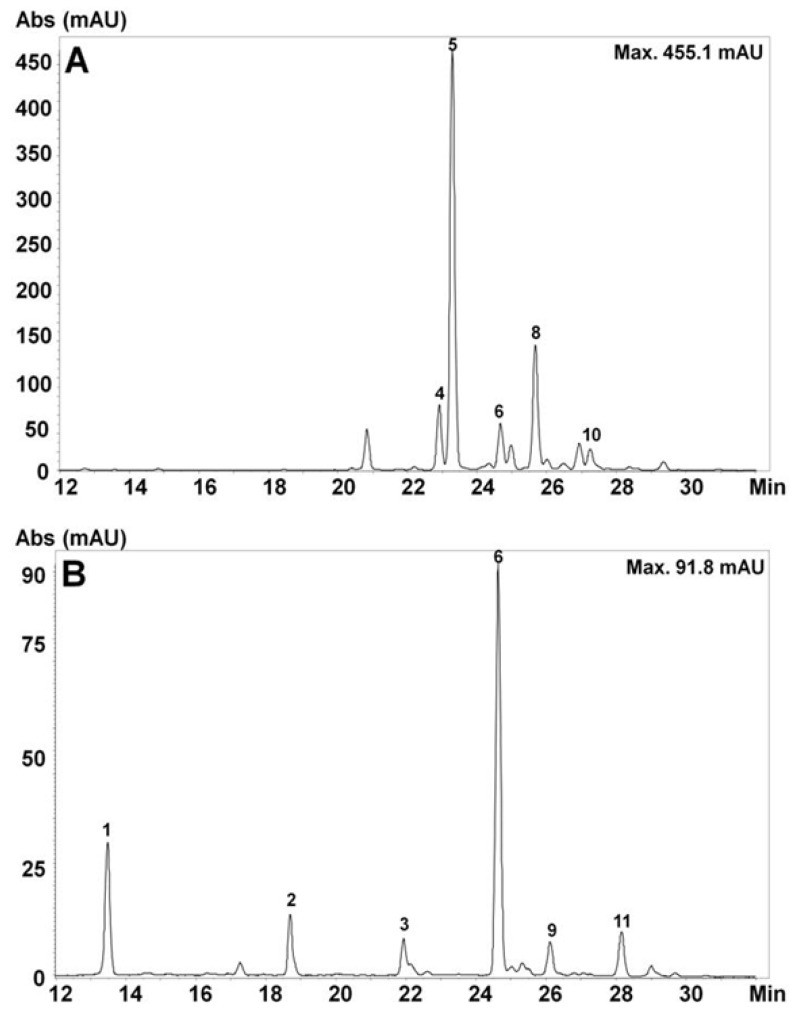
High-performance liquid chromatography diode array detection (HPLC-DAD) chromatograms of anthocyanins of fresh colored fleshed potatoes from southern Chile (520 nm). Identity assignment of anthocyanins according to Table 1. Where: (**A**) purple-fleshed potatoes (CB2011.104), (**B**) red-fleshed potatoes (TR2012.078).

**Figure 2 molecules-26-00314-f002:**
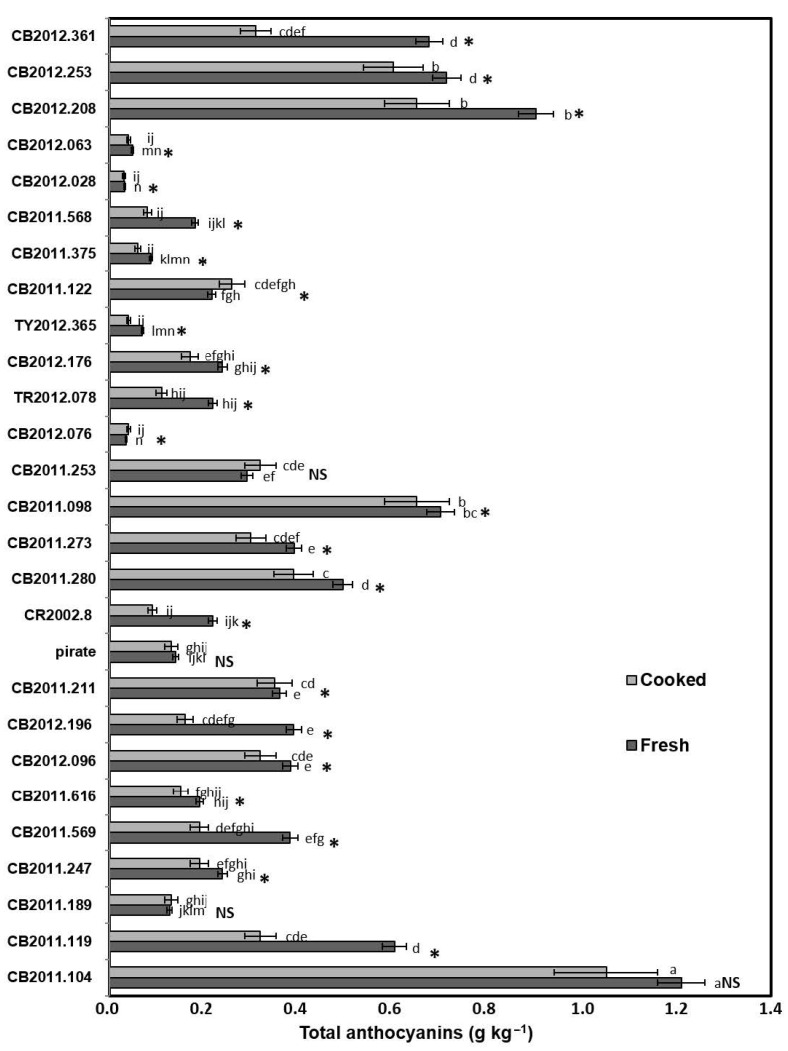
Effects of the cooking process in concentrations of anthocyanins (g kg^−1^ fresh weight) by HPLC-DAD (520 nm) in colored fleshed potatoes from Southern Chile. Different letters in each bar indicate significant differences according to Tukey’s multiple range test (*p* < 0.05). For each genotype, asterisks (*) indicate differences between fresh and cooked samples according to the Student *t*-test (*p* < 0.05).

**Figure 3 molecules-26-00314-f003:**
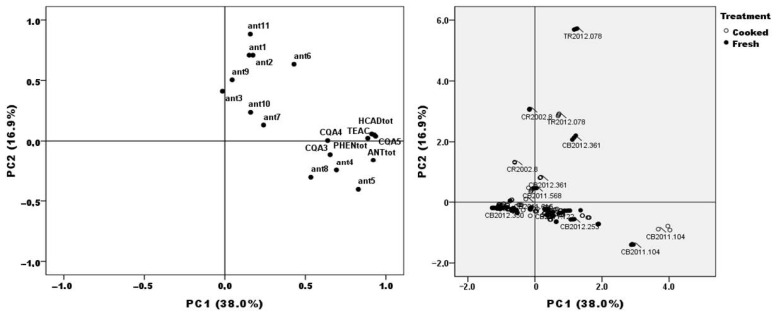
Principal component (PC) analysis for individual and total anthocyanin and hydroxycinnamic acid derivatives concentrations, antioxidant activity (TEAC), and total phenols (Folin–Ciocalteau) of all genotypes used in this study. Percentage values in parentheses indicate the experimental variation explained by each PC. **Left**: principal component analysis (PCA) distribution of the diverse experimental variables here analyzed based in its loading values. **Right:** Distribution of all the experimental units according to the scores regarding the two first PC extracted.

**Figure 4 molecules-26-00314-f004:**
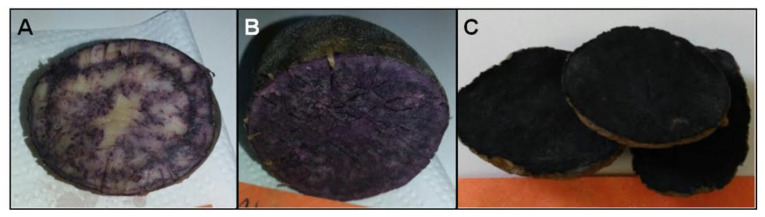
Some purple-fleshed potato genotypes were used in this study. (**A**) Low colored genotype CB2011.098. (**B**) Medium colored genotype CB2011.208. (**C**) High colored genotype CB2011.104.

**Table 1 molecules-26-00314-t001:** Identification of anthocyanins from colored potatoes by a liquid chromatography diode array detector coupled to a mass spectrometer with an electrospray ionization interface (HPLC-DAD-ESI-MS/MS).

Peak	t_R_ (min)	Identity Assignment	λ (nm)	M^+^	Product Ions
1	13.48	pelargonidin-3-caffeoylrutinoside	500, 420 (sh), 326, 275	741.2	271.1, 579.1, 433.1
2	18.69	pelargonidin-3-rutinoside	504, 430 (sh), 320 (sh), 275	579.1	271.3, 433.1
3	21.93	pelargonidin-3-caffeoylrutinoside-5-glucoside	509, 430 (sh), 314, 289	903.2	271.2, 741.2, 433.1
4	22.92	cyanidin-3-*p*-coumaroylrutinoside-5-glucoside	521, 311, 292, 281	903.4	449.2, 287
5	23.29	petunidin-3-*p*-coumaroylrutinoside-5-glucoside	530, 306, 279	933.2	771.2, 479.0, 317.0
6	24.64	pelargonidin-3-*p*-coumaroylrutinoside-5-glucoside	504, 425 (sh), 313, 286	887.3	271.3, 725.2, 433.1
7	24.67	petunidin-3-feruloylrutinoside-5-glucoside	531, 326, 298, 279	963.3	479.0, 317.0
8	25.67	peonidin-3-*p*-coumaroylrutinoside-5-glucoside	521, 311, 293, 280	917.2	755.3, 463.2, 301.2
9	26.12	pelargonidin-3-feruloylrutinoside-5-glucoside	504, 420 (sh), 323, 286	917.3	271.1 755.2, 433.1
10	27.25	malvidin-3-*p*-coumaroylrutinoside-5-glucoside	520, 323, 290, 280	947.4	
11	28.16	pelargonidin-3-*p*-coumaroylrutinoside-5-formylglucoside	504, 314, 286	915.1	725.5, 461.3, 271.1

**Table 2 molecules-26-00314-t002:** Antioxidant activity (trolox equivalent antioxidant capacity (TEAC)) and total phenols (Folin–Ciocalteau method) in fresh and cooked white and colored potatoes. All concentrations are expressed as fresh weight.

	TEAC (µmol g^−1^ Trolox Equivalents)			Total Phenols (g kg^−1^)		
Sample	Fresh	Cooked		Fresh	Cooked	
CB2011.104	8.35 ± 0.42 ^a^	8.07 ± 0.40 ^a^	*	2.42 ± 0.05 ^a^	2.34 ± 0.05 ^a^	*
CB2011.119	2.15 ± 0.11 ^ghij^	4.65 ± 0.23 ^bc^	*	0.65 ± 0.01 ^kl^	0.99 ± 0.02 ^e^	*
CB2011.189	1.08 ± 0.05 ^klmn^	3.31 ± 0.17 ^def^	*	0.85 ± 0.02 ^lm^	0.71 ± 0.01 ^fg^	NS
CB2011.247	1.37 ± 0.07 ^kln^	3.41 ± 0.17 ^def^	*	1.19 ± 0.02 ^i^	0.73 ± 0.01 ^f^	*
CB2011.569	1.98 ± 0.10 ^jkl^	3.09 ± 0.15 ^efg^	*	0.79 ± 0.02 ^no^	0.45 ± 0.01 ^jk^	*
CB2011.616	3.48 ± 0.17 ^hij^	3.13 ± 0.16 ^efg^	*	1.00 ± 0.02 ^klm^	0.62 ± 0.01 ^gh^	*
CB2012.096	2.54 ± 0.13 ^hij^	3.97 ± 0.20 ^cde^	NS	1.46 ± 0.03 ^ij^	0.33 ± 0.01 ^lm^	*
CB2012.196	4.91 ± 0.25 ^defgh^	3.32 ± 0.17 ^def^	*	1.38 ± 0.03 ^g^	1.13 ± 0.02 ^d^	*
CB2011.211	3.91 ± 0.20 ^defg^	4.48 ± 0.22 ^bc^	NS	1.55 ± 0.03 ^n^	0.94 ± 0.02 ^e^	*
BWF	0.85 ± 0.04 ^grs^	1.01 ± 0.05 ^jk^	NS	0.61 ± 0.01 ^no^	0.55 ± 0.01 ^hi^	NS
pirate	1.12 ± 0.06 ^opqr^	1.45 ± 0.07 ^hij^	NS	0.64 ± 0.01 ^no^	0.50 ± 0.01 ^ijk^	*
VR808	0.54 ± 0.03 ^rs^	1.06 ± 0.05 ^jk^	NS	0.21 ± 0.00 ^v^	0.21 ± 0.00 ^n^	NS
CR2002.8	2.60 ± 0.13 ^klmn^	2.20 ± 0.11 ^ghi^	NS	0.79 ± 0.02 ^n^	0.48 ± 0.01 ^ijk^	*
CB2011.280	2.85 ± 0.14 ^ijk^	2.85 ± 0.14 ^fg^	NS	1.23 ± 0.02 ^n^	1.02 ± 0.02 ^e^	*
CB2011.273	3.17 ± 0.16 ^fghi^	4.12 ± 0.21 ^cd^	NS	1.92 ± 0.04 ^d^	1.30 ± 0.03 ^c^	*
CB2011.098	4.60 ± 0.23 ^cd^	5.80 ± 0.29 ^b^	*	1.81 ± 0.04 ^c^	1.70 ± 0.03 ^b^	NS
CB2011.253	4.63 ± 0.23 ^fghi^	3.94 ± 0.20 ^cde^	*	1.59 ± 0.03 ^f^	1.31 ± 0.03 ^c^	*
CB2012.076	1.90 ± 0.09 ^mnopqr^	1.18 ± 0.06 ^j^	*	0.69 ± 0.01 ^no^	0.51 ± 0.01 ^ij^	*
TR2012.078	5.19 ± 0.26 ^c^	5.27 ± 0.26 ^b^	NS	1.68 ± 0.03 ^f^	1.02 ± 0.02 ^e^	*
CB2012.176	3.94 ± 0.20 ^efgh^	3.51 ± 0.18 ^def^	*	0.92 ± 0.02 ^ij^	0.93 ± 0.02 ^e^	NS
CB2012.347	0.45 ± 0.02 ^s^	nd ^k^	*	0.27 ± 0.01 ^v^	0.26 ± 0.01 ^mn^	NS
TY2012.365	1.93 ± 0.10 ^mnopqr^	1.23 ± 0.06 ^ij^	*	0.40 ± 0.01 ^q^	0.29 ± 0.01 ^mn^	*
CB2011.122	4.14 ± 0.21 ^efgh^	3.87 ± 0.19 ^cde^	NS	1.12 ± 0.02 ^n^	1.14 ± 0.02 ^d^	NS
CB2011.375	2.04 ± 0.10 ^lmnopq^	1.57 ± 0.08 ^hij^	*	0.63 ± 0.01 ^o^	0.49 ± 0.01 ^ijk^	*
CB2011.568	3.18 ± 0.16 ^klmno^	1.09 ± 0.05 ^j^	*	1.22 ± 0.02 ^jl^	0.50 ± 0.01 ^ijk^	*
CB2012.028	2.18 ± 0.11 ^klmnop^	1.84 ± 0.09 ^hij^	NS	0.54 ± 0.01 ^p^	0.31 ± 0.01 ^m^	*
CB2012.063	1.31 ± 0.07 ^nopqr^	1.72 ± 0.09 ^hij^	NS	0.48 ± 0.01 ^p^	0.41 ± 0.01 ^kl^	NS
CB2012.128	0.72 ± 0.04 ^nopqr^	2.31 ± 0.12 ^gh^	*	0.78 ± 0.02 ^m^	0.70 ± 0.01 ^fg^	NS
CB2012.208	6.76 ± 0.34 ^b^	7.80 ± 0.39 ^a^	NS	2.23 ± 0.04 ^b^	1.78 ± 0.04 ^b^	*
CB2012.253	3.51 ± 0.10 ^def^	3.54 ± 0.23 ^def^	NS	1.96 ± 0.04 ^g^	0.58 ± 0.01 ^hi^	*
CB2012.350	1.11 ± 0.06 ^grs^	1.00 ± 0.05 ^jk^	NS	0.45 ± 0.01 ^pq^	0.32 ± 0.01 ^lm^	*
CB2012.361	5.39 ± 0.27 ^cde^	3.69 ± 0.18 ^cdef^	*	1.97 ± 0.04 ^e^	0.98 ± 0.02 ^e^	*
CR2012.363	1.38 ± 0.07 ^pgr^	1.06 ± 0.05 ^jk^	*	0.48 ± 0.01 ^pq^	0.29 ± 0.01 ^mn^	*

Different letters in a column indicate significant differences according to Tukey’s multiple range test (*p* < 0.05). For each genotype, asterisks (*) indicate differences between fresh and cooked samples according to the Student *t*-test (*p* < 0.05).

## Data Availability

The data presented in this study are available on request from the corresponding author.

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
