# Peer review of "Noticeable Quantities of Functional Compounds and Antioxidant Activities Remain after Cooking of Colored Fleshed Potatoes Native from Southern Chile"

_molecules, 2021, doi:10.3390/molecules26020314_

Round 1
Reviewer 1 Report
The authors improved their work.
Reviewer 2 Report
The authors, following the suggestions of the reviewers, are proving a new version of manuscript in which text and content have been strongly improved.
I have only one request concerning PCA figure: may the authors provide a new figure with a better quality? For example, one figure in which the two panels have the same dimensions.
This manuscript is a resubmission of an earlier submission. The following is a list of the peer review reports and author responses from that submission.
Round 1
Reviewer 1 Report
The authors investigated the chemical profile of the colored fleshed potatoes using LC-MS. However, the manuscript is very weak and needs a lot of improvements.
- There are many typos, please revise the manuscript thoroughly
- The identification of the compounds is poor since it depends solely on MS data. Surprisingly, in many cases, the authors mention a certain sugar, although MS cannot distinguish between different sugars. In addition, no MS spectra are shown (should be added in the Suppl. Data). The authors must either compare the compounds with authentic standards (MS and retention time) or use additional techniques (NMR, IR, CD spectroscopy etc.) to confirm identity.
- The authors represented the identified compounds but no information about which genotype; the authors should provide a comprehensive catalog of the secondary metabolites in each genotype and quantify it and then compare between the genotypes using a chemometric approach.
- The extraction methods are not clear, elaborate.
- The quantification procedure is not clear. How did you quantify anthocyanins in the cooked potatoes? Have you taken into consideration the water content of the potatoes after cooking?
Reviewer 2 Report
The manuscript entitled “Noticeable quantities of functional compounds and antioxidant activities remain after cooking of colored fleshed potatoes native from Southern Chile” authored by Stefano Ercoli and colleagues, deals with the investigation of the phytochemical profile and with the evaluation of the antioxidant activity of in 33 colored-fleshed potatoes genotypes, before and after cooking process. The purpose of the work, and the results obtained are original and certainly interesting. However, before considering it acceptable as a publication in Molecules, some considerations need to be made:
INTRODUCTION: the introduction is well written. Surely, something more could be mentioned regarding the limited distribution of anthocyanins in plant sources. Some information can be found in these papers: https://doi.org/10.3390/nu12040992, https://doi.org/10.1016/j.foodchem.2020.127439, and https://doi.org/10.3389/fnut.2019.00133
LINE 38-40: this sentence is not properly true. Among the vegetables, or rather among the tubers, potato can be considered one of the sources with the highest content of polyphenols. However, many other sources, mainly fruits, have a polyphenolic content even 100-fold higher. Please, modify the sentence accordingly. I also recommend using a better biographical source (ex. https://doi.org/10.1038/ejcn.2010.221).
MATERIALS AND METHODS SECTION: the materials and methods section is incomplete. In order to improve the quality of this section, I suggest the authors to carefully follow these few guidelines.
- In subsection 4.2. it is necessary to report the chromatographic gradient used to separate the individual phenolic compounds, the type of column and other conditions necessary to replicate the analytical analyses. I understand that part of this information is present in the references provided by the authors, but it is necessary to minimally report it.
- Moreover, no information regarding the validation of the extraction and analytical method is present in this subsection. I wonder if the authors evaluated the completeness of their extraction processes.
- Therefore, I strongly recommend the inclusion of the analytical method validation, in which not only the recovery and linearity of calibration curves should be added, but also information regarding matrix effect (ME), limit of detections (LOD), limit of quantification (LOQ), and precision. Authors can refer to this paper, where the same parameters were evaluated for anthocyanins from cranberry samples (https://doi.org/10.3390/nu12040992).
- The description of the quantification of the total phenolic compounds should be placed in an additional paragraph. Also in this case, I understand that the method is well known, but at the same time it is one of the most personalized methods among laboratory practices. Consequently, report a small description of the method accordingly.
- Please, provide some information about TAEC methodology in Subsection 4.3.
RESULTS: anthocyanins are certainly the most interesting compounds among plant matrices, both thanks to their limited distribution and to their powerful antioxidant activity. In this work, the authors did not limit their attention only on the identification and quantification of anthocyanins, but they also investigated other phenolic acids, including hydroxycinnamic acids. However, this last aspect does not immediately stand out, probably due to the lack of a table reporting the list of these compounds. I would therefore suggest to include also these compounds in Table 1 (by changing the table caption properly) or to add in the manuscript an additional table reporting these compounds.
LINE 77: please, remove the dot before “2”.
LINE 83-84: authors cannot write this ruling unless they first evaluate LOD and LOQ.
DISCUSSION: the discussion is well written, and follows a logical thread. However, the authors have not minimally compared the data obtained in this work with those previously reported in the literature on the same raw material. Authors can use some of these article: https://doi.org/10.1177/1934578X20936931; https://doi.org/10.3390/foods9111598; https://doi.org/10.1007/s12230-018-09703-6, https://doi.org/10.3390/plants9070815.
REFERENCES: The last problem regard the reference list. The articles cited by the authors are quite limited (altogether 30), and many of these have not been published in the last 5 years. Taking into consideration the clarifications made in this reports, I would suggest to introduce more recent references. For example, those reported in this peer review were published in 2019-2020.